# CRISPRi-Guided Metabolic Flux Engineering for Enhanced Protopanaxadiol Production in *Saccharomyces cerevisiae*

**DOI:** 10.3390/ijms222111836

**Published:** 2021-10-31

**Authors:** Soo-Hwan Lim, Jong-In Baek, Byeong-Min Jeon, Jung-Woo Seo, Min-Sung Kim, Ji-Young Byun, Soo-Hoon Park, Su-Jin Kim, Ju-Young Lee, Jun-Hyoung Lee, Sun-Chang Kim

**Affiliations:** 1Intelligent Synthetic Biology Center, 291 Daehak-ro, Daejeon 305-701, Korea; lshbio@kaist.ac.kr (S.-H.L.); jwseo123@kaist.ac.kr (J.-W.S.); shpark11@guest.kaist.ac.kr (S.-H.P.); food7568@gmail.com (S.-J.K.); junhlee@kaist.ac.kr (J.-H.L.); 2Department of Biological Sciences, Korea Advanced Institute of Science and Technology, 291 Daehak-ro, Daejeon 305-701, Korea; baekji@kaist.ac.kr (J.-I.B.); jbm0901@kaist.ac.kr (B.-M.J.); mskim8906@kaist.ac.kr (M.-S.K.); byunjy@kaist.ac.kr (J.-Y.B.); 3Research Center for Bio-Based Chemistry, Korea Research Institute of Chemical Technology (KRICT), 406-30, Jongga-ro, Ulsan 44429, Korea; juylee@krict.re.kr; 4KAIST Institute for Biocentury, Korea Advanced Institute of Science and Technology, 291 Daehak-ro, Daejeon 305-701, Korea

**Keywords:** CRISPR interference, lanosterol, protopanaxadiol, triterpenes, metabolic engineering, ginsenosides, *Saccharomyces cerevisiae*

## Abstract

Protopanaxadiol (PPD), an aglycon found in several dammarene-type ginsenosides, has high potency as a pharmaceutical. Nevertheless, application of these ginsenosides has been limited because of the high production cost due to the rare content of PPD in *Panax ginseng* and a long cultivation time (4–6 years). For the biological mass production of the PPD, de novo biosynthetic pathways for PPD were introduced in *Saccharomyces cerevisiae* and the metabolic flux toward the target molecule was restructured to avoid competition for carbon sources between native metabolic pathways and de novo biosynthetic pathways producing PPD in *S. cerevisiae*. Here, we report a CRISPRi (clustered regularly interspaced short palindromic repeats interference)-based customized metabolic flux system which downregulates the lanosterol (a competing metabolite of dammarenediol-II (DD-II)) synthase in *S. cerevisiae*. With the CRISPRi-mediated suppression of lanosterol synthase and diversion of lanosterol to DD-II and PPD in *S. cerevisiae*, we increased PPD production 14.4-fold in shake-flask fermentation and 5.7-fold in a long-term batch-fed fermentation.

## 1. Introduction

Protopanaxadiol (PPD), an aglycon of dammarene-type ginsenosides, possesses anti-cancer, anti-inflammatory, anti-oxidant, hepatoprotectant, anti-lipogenic, wound-healing, and anti-obesity activity [1,2,3,4,5]. Despite having high pharmacological potency, the application of these ginsenosides as drugs and pharmaceuticals is limited because *Panax ginseng* cultivation is slow, and the ginsenoside content of ginseng is low [6]. To overcome this, PPD and ginsenosides have been produced using de novo biosynthetic pathways in heterologous microbial organisms, including yeast [7,8,9,10,11,12]. PPD is biosynthesized from 2,3-oxidosqualene, which is derived from isopentenyl diphosphate and dimethylallyl diphosphate ([13]. These precursors are synthesized via the mevalonate (MVA) pathway in both ginseng and yeast [14]. PPD biosynthesis from 2,3-oxidosqualene in yeast requires three ginseng-derived enzymes—*P. ginseng* dammarenediol-II (DD-II) synthase (PgDS), *P. ginseng* PPD synthase (PgPPDS), and *P. ginseng* cytochrome P450 reductase (PgCPR1) [15,16,17]. PgDS, an oxidosqualene cyclase, converts 2,3-oxidosqualene to DD-II. PgPPDS and PgCPR1 together catalyze the hydroxylation of DD-II at the C-12 position, resulting in PPD biosynthesis. PPD then undergoes glycosylation catalyzed by several UDP-glucosyltransferases, yielding ginsenosides [13].

In yeast, however, 2,3-oxidosqualene is natively converted to lanosterol, catalyzed by lanosterol synthase, and then used for biosynthesis of sterols such as ergosterol [18]. Thus, in yeast, the PPD biosynthetic pathway competes with the native sterol biosynthetic pathway for 2,3-oxidosqualene. Since deletion of the *ERG7* gene, which encodes lanosterol synthase in yeast, is lethal, it is not possible to completely block the sterol biosynthetic pathway [18]. Rather, it is necessary to quantitatively control *ERG7* expression (i.e., partial suppression) in order to enhance the PPD biosynthetic pathway. For example, in *Saccharomyces cerevisiae*, mass production of heterologous terpenes, which are synthesized from MVA, requires partial suppression of the metabolic flux of native sterol biosynthesis, in order to increase the metabolic flux of the target terpenes; this is achieved via replacement of the native promoter with the methionine-repressible *MET3* promoter, the copper-repressible *CTR3* promoter, or a constitutively weak promoter, or via fusion of the target enzyme to a degradation tag [19,20,21,22,23,24,25,26]. However, this strategy has several limitations. First, adding methionine increases production costs because it is consumed by yeast. Second, adding copper can result in toxicity in yeast. Third, it is difficult to quantitatively control the metabolic flux by replacing the native promoter with a constitutively weak promoter; further, selection of a suitable promoters is labor-intensive and time-consuming [27,28,29,30].

To address this, we developed a CRISPRi-based PPD biosynthesis approach for yeast. We constructed a PPD-producing yeast strain and engineered it to respond to increasing levels of the precursors. This approach alters cellular carbon flux and partially suppresses competing metabolic pathways to improve PPD production in yeast. 

## 2. Results

### 2.1. Construction of the PPD-Producing Yeast Strain

Because dammarenediol-II synthase (DS), protopanaxadiol synthase (PPDS), and cytochrome P450 reductase (CPR) are necessary to construct the PPD-producing yeast strain, we first constructed *PgDS* and *PgPPDS* expression cassettes that are controlled by the *GPD* promoter and a *PgCPR* expression cassette that is controlled by the *PGK1* promoter (Figure 1); these three cassettes were then integrated into the CEN.PK2-1D strain, producing the PPD-A1 strain (Table 1).

To evaluate PPD production in the PPD-A1 strain, we performed shake-flask fermentation using strains CEN.PK2-1D and PPD-A1 in YPD medium for 48 h (Figure 2). In the CEN.PK2-1D strain, squalene, 2,3-oxidosqualene, lanosterol, and ergosterol were produced at 21.3, 2.8, 9.9, and 49.7 mg/L, respectively. In the PPD-A1 strain, PPD was produced at 1.5 mg/L. DD-II was not accumulated, suggesting that it was completely converted to PPD. The amounts of lanosterol and ergosterol produced were similar in both strains. These results verify that the PPD-A1 strain produced PPD.

### 2.2. Enhancing the MVA Pathway and Squalene Monooxygenase Stability Upregulated the Triterpene Biosynthetic Pathway

Enhancing MVA-pathway metabolic flux is an efficient way to improve triterpene production in yeast [7,11,31,32,33]. We used tHMGR1-overexpression to improve PPD production (Figure 1). We constructed the tHMGR1 expression cassette controlled by the *GPD* promoter and integrated it into the PPD-A1 strain, producing the PPD-A2 strain (Table 1), which was then subjected to shake-flask fermentation. However, the PPD production of PPD-A2 was not better than that of PPD-A1, although PPD-A2 produced substantially more squalene, at 564.4 mg/L (Figure 2). PPD-A2 produced more lanosterol, at 38.6 mg/L, but did not differ from PPD-A1 in 2,3-oxidosqualene and ergosterol production. Therefore, tHMGR1-overexpression effectively increased the production of squalene and lanosterol, but not of PPD.

To convert more of the accumulated squalene into 2,3-oxidosqualene, it is necessary to overexpress *ERG1*. *ERG1* is a key regulator of sterol homeostasis [18,34]. In yeast, excess lanosterol accumulation leads to *ERG1* degradation via the ER-associated protein degradation (ERAD) pathway [34] found that *ERG1* was a target of the ERAD-associated protein ubiquitin ligase Doa10, whereas an *ERG1* derivate (K278R/K284R/K311R/K360R) was stabilized from ERAD-mediated protein degradation. Here, we chose to overexpress this *ERG1*-derivate (*ERG1m* hereafter), rather than the native *ERG1*, in the PPD-A2 strain. To do this, we constructed the *ERG1m* expression cassette controlled by the *TEF1* promoter and integrated it into the PPD-A2 strain, producing the PPD-A3 strain. Following shake-flask fermentation of PPD-A3, squalene production was indeed lower in PPD-A3, at 96.1 mg/L, than in PPD-A2 (Figure 2), and PPD production was higher, at 2.2 mg/L. Notably, lanosterol production was also higher in PPD-A3, at 265.8 mg/L, whereas that of ergosterol was similar, relative to that of PPD-A2. Therefore, enhancing the MVA pathway via tHMGR1 overexpression and increasing *ERG1* stability via *ERG1m* overexpression improved the triterpene biosynthetic pathway and particularly, lanosterol and PPD production.

### 2.3. CRISPRi-Mediated ERG7 Suppression Improved PPD Production

Although PPD-A3 produced more PPD than PPD-A2, lanosterol showed greater fold change than PPD from PPD-A2 to PPD-A3, suggesting that, in PPD-A3, most of the 2,3-oxidosqualene was converted to lanosterol rather than DD-II and PPD. To further improve PPD production in engineered yeast strains, it is necessary to efficiently suppress lanosterol biosynthesis by quantitative suppression of *ERG7* expression. Partial suppression of *ERG7* expression using the methionine-suppressible *MET3* promoter or antisense *ERG7* fragment has been reported [23,31,35]. However, *MET3* promoter-mediated *ERG7* suppression requires extra methionine to continuously suppress *ERG7* expression. Further, antisense *ERG7* fragment-mediated *ERG7* suppression did not quantitatively suppress *ERG7* expression. In addition, the long antisense RNA involved might be unstable. To address these limitations, we used the CRISPRi system to constitutively and quantitatively suppress *ERG7* expression. We first designed five *ERG7* promoter-targeting sgRNAs (sgRNA1–5) using an online tool (https://lp2.github.io/yeast-crispri/, accessed 1 December 2017) (Figure 3). We then synthesized five *SNR52*pro-sgRNA-*SUP4*ter cassettes and cloned them into the dCas9-expressing plasmid pTDH3-dCas9-Mxi1. To co-express the dCas9 and sgRNA in yeast, we constructed five dCas9-sgRNA cassettes and integrated them into the PPD-A3 strain, producing strains PPD-A3-sgRNA1–5. To determine whether the dCas9-sgRNA cassettes suppress *ERG7* expression and improve PPD production in yeast, we performed shake-flask fermentation of strains PPD-A3 and PPD-A3-sgRNA1–5 in YPD medium for 48 h (Figure 3 and Figure 4). We first analyzed the *ERG7* expression levels of these strains using quantitative RT-PCR (Figure 3). Relative to the expression in the control strain (PPD-A3), the *ERG7* expression levels of strains PPD-A3-sgRNA1–5 were decreased 0.73-, 0.58-, 0.24-, 0.40-, and 0.66-fold, respectively. PPD-A3-sgRNA3 exhibited the most effective *ERG7* suppression. To exclude the possibility that dCas9 expression differed among strains PPD-A3-sgRNA1–5, we analyzed their dCas9 expression using quantitative RT-PCR (Appendix A); dCas9 expression was similar among these strains, suggesting that their different *ERG7* expression resulted from different sgRNA efficiencies.

The cell growth of PPD-A3-sgRNA1–5 was lower than that of PPD-A3 (Figure 4A and Appendix A). In particular, PPD-A3-sgRNA3 exhibited severe growth retardation, suggesting that excessive *ERG7* suppression might damage cell viability. To determine whether *ERG7* suppression by dCas9-sgRNAs resulted in the suppression of lanosterol biosynthesis and improvement of PPD production, we analyzed lanosterol and PPD production after shake-flask fermentation, using the PPD-A3 and PPD-A3-sgRNA1–5 strains (Figure 4B). Lanosterol production in PPD-A3 was 258.6 mg/L, and was lower, at 198.2, 165.9, 44.8, 152.7, and 200.8 mg/L, respectively, in PPD-A3-sgRNA1–5. Notably, the order of the suppression efficiency of lanosterol biosynthesis in strains PPD-A3-sgRNA1–5 was the same as that for *ERG7* expression, suggesting that dCas9-sgRNA could quantitatively suppress both *ERG7* expression and lanosterol biosynthesis. PPD production by PPD-A3 was 1.9 mg/L, and was higher, at 11.2, 23.0, 17.8, 27.6, and 11.1 mg/L, respectively, in PPD-A3-sgRNA1–5. Notably, PPD-A3-sgRNA3, which exhibited the strongest *ERG7* suppression, did not exhibit the highest PPD production. Rather, PPD-A3-sgRNA4 exhibited the highest PPD production, suggesting that excessive *ERG7* suppression might disadvantage the biosynthetic pathway of the secondary metabolite. This indicates that our method efficiently suppressed *ERG7* expression, thereby improving PPD production in the engineered yeast strain.

### 2.4. PPD Production via Batch-Fed Fermentation

Although several studies have assessed the use of CRISPRi to regulate metabolic flux in yeast, most of them have been limited to a few days in duration [36,37,38,39]. To determine whether dCas9-sgRNA-mediated *ERG7* suppression was effective over longer periods, we performed batch-fed fermentation, using the PPD-A3 and PPD-A3-sgRNA4 strains, in a 5-L bioreactor for 216 h; the strains exhibited similar cell growth profiles (Figure 5A). Lanosterol production by PPD-A3 reached 541.5 mg/L at 120 h and maintained this until 216 h (Figure 5B). For PPD-A3-sgRNA4, however, lanosterol production reached 210 mg/L at 120 h, then decreased to 140 mg/L at 216 h (Figure 5B). At 216 h, PPD production by PPD-A3 and PPD-A3-sgRNA4 was 52.1 mg/L and 294.5 mg/L, respectively (Figure 5C). This indicates that dCas9-sgRNA-mediated *ERG7* suppression over longer periods effectively suppressed lanosterol biosynthesis and improved PPD production in the engineered strain.

## 3. Discussion

PPD has important pharmacological properties, including anti-cancer activity. To improve production of this high-value-added product, we metabolically engineered yeast strains to biosynthesize it more efficiently. We first constructed a PPD-producing yeast strain in which tHMGR1 was overexpressed to enhance the MVA pathway. Next, we overexpressed the *ERG1m* gene in this engineered strain, to enhance squalene conversion to 2,3-oxidosqualene. In spite of this, PPD production remained low, because the PPD and native ergosterol biosynthetic pathways compete for 2,3-oxidosqualene. To overcome this competition, we established a CRISPRi-based metabolic engineering strategy, and achieved stable, sustained, high-level PPD production in *S. cerevisiae*.

In *S. cerevisiae*, HMGR1 and *ERG1* are key regulatory enzymes in the sterol biosynthetic pathway [34,40]. They are regulated by negative feedback at transcriptional and protein levels, to maintain ergosterol homeostasis. When we overexpressed tHMGR1 in the PPD-producing yeast strain, squalene accumulated, whereas 2,3-oxidosqualene did not accumulate, suggesting that *ERG1* is also a rate-limiting enzyme in the sterol biosynthetic pathway. When we further overexpressed the *ERG1m* gene in the PPD-producing strain and most of the squalene was converted to lanosterol and PPD. However, lanosterol production far exceeded PPD production, suggesting that most of the 2,3-oxidosqualene were converted to lanosterol rather than DD-II and PPD. We propose a possible reason for this—the enzymatic activity of *ERG7* might be stronger than that of PgDS. In humans, lanosterol synthase is a monotopic endoplasmic reticulum (ER)-membrane protein comprising two (α/α) barrel domains connected by loops, and three smaller β-structures [41]. The active site cavity is located in the center of the enzyme, and 2,3-oxidosqualene enters this active site cavity. Although PgDS also functions at the ER, it does not contain a transmembrane domain [42]. Thus, *ERG7* might have a competitive advantage over PgDS in substrate binding. To overcome this competition, we decided to suppress the *ERG7* expression in engineered yeast strains.

To suppress *ERG7* expression, we applied CRISPRi-guided regulation of *ERG7* expression, using dCas9 and five *ERG7* promoter-targeting sgRNAs (sgRNA1–5). Since the stable expression of dCas9 and sgRNA is important for constitutive suppression of target gene expression, we integrated dCas9-sgRNA cassettes into chromosomes of the PPD-A3 strain. dCas9 expression was similar among these strains, whereas *ERG7* expression differed substantially. The efficiency of the sgRNAs that we constructed for *ERG7* suppression was ranked (in decreasing order), as follows: sgRNA3, sgRNA4, sgRNA2, sgRNA5, and sgRNA1. Notably, the dCas9-sgRNA3 cassette retarded cell growth, suggesting that there might be an *ERG7* expression threshold that affects cell viability. It is possible, by using different sgRNAs targeting the same gene, to reveal differences in efficiency [37,38,39,42]. It is likely that sgRNA3 was the most effective at suppressing *ERG7* expression because its target region is closest to the TATA-box in the *ERG7* promoter. Indeed, sgRNA1 and sgRNA5, for which the target regions are distant from the TATA-box in the *ERG7* promoter, were less effective at suppressing *ERG7* expression. These findings indicate that the sgRNA target site is a key aspect in the quantitative regulation of gene expression. Further, with the exception of *ERG7* suppression by sgRNA3, the relative differences in the efficiency of the constructed sgRNAs were consistent with those in PPD production. This indicates that CRISPRi-guided suppression of *ERG7* expression suppressed lanosterol biosynthesis and improved PPD production.

Notably, CRISPRi-guided suppression of *ERG7* resulted in the accumulation of squalene and 2,3-oxidosqualene, but not ergosterol. Yeast cells not only regulate sterol biosynthesis but also convert excessive sterols into steryl esters, which are then stored in lipid droplets or secreted into the extracellular matrix [18]. Considering that overexpression of *tHMGR1* and *EGR1m* enhanced sterol biosynthesis by inhibiting negative feedback, accumulation of squalene and 2,3-oxidosqualene indicates that CRISPRi-guided *ERG7* suppression is a powerful tool for regulating metabolic flux. To further enhance the conversion of 2,3-oxidosqualene to DD-II and PPD in yeast, it might be helpful to increase *PgDS*, *PgPPDS*, and *PgCPR* expression.

The stability of the dCas9-sgRNA complex, and the accuracy of sgRNA-targeting, are important factors for achieving stable and efficient regulation of target gene expression over longer periods, when using CRISPRi to bioengineer yeast strains. Most previous studies on this have been conducted over a few days, possibly because of difficulties in maintaining dCas9-sgRNA complex stability [37,38,39,42]. Here, we performed batch-fed fermentation over 9 days, using the PPD-A3 and PPD-A3-sgRNA4 strains, to confirm the effect of CRISPRi-guided *ERG7* suppression over longer periods. At 216 h, lanosterol production was lower, but PPD production was higher in the PPD-A3-sgRNA4 strain than in the unmodified PPD-producing (control) strain. This is the first study to demonstrate the efficacy of CRISPRi-guided regulation of metabolic flux for longer periods. These findings may help in the metabolic engineering of industrial strains to improve production of target molecules.

## 4. Materials and Methods

### 4.1. Strains and Medium

*Saccharomyces cerevisiae* CEN.PK2-1D was obtained from EUROSCARF (http://www.euroscarf.de/) and used as a parent strain for all engineered yeast strains. The yeast strains are listed in Table 1. Constructed yeast strains were grown in YPD medium (10 g/L yeast extract, 20 g/L peptone, and 50 g/L glucose) or SD medium (6.7 g/L yeast nitrogen base and 50 g/L glucose) at 26 °C, with shaking at 220 rpm, lacking histidine, tryptophan, leucine, and uracil where appropriate. 

*Escherichia coli* DH5α (Enzynomics, Republic of Korea) was used for transformation and plasmid amplification. *Escherichia coli* cells were grown in Luria–Bertani (LB) medium (10 g/L tryptone, 5 g/L yeast extract, and 10 g/L NaCl) with 100 mg/L ampicillin at 37 °C, with shaking at 200 rpm. 

### 4.2. Plasmid Construction

Yeast expression plasmids pRS424TEF1(#87365), pRS426GPD(#87361), pRS426PGK1(#1370), Cas9-NAT(#64329), and pTDH3-dCas9-Mxi1(#46921) and sgRNA-expressing plasmid pRS42H(#64330) were purchased from Addgene (Cambridge, MA, USA), and sgRNA-expressing plasmid pRS42K(P30637) was purchased from EUROSCARF (Frankfurt, Germany). Dammarenediol-II synthase (*PgDS*; GenBank: AB265170), PPD synthase (*PgPPDS*; GenBank: JN604537), and cytochrome P450 reductase (*PgCPR*; GenBank: KF486915), each with the BamHI and XhoI restriction sites at their 5′- and 3′- ends, respectively, were synthesized with codon optimization for expression in *S. cerevisiae* (BIONEER, Daejeon, Korea). *PgDS* and *PgPPDS* were cloned into the pRS426GPD vector, and *PgCPR* was cloned into pRS426PGK1. For the CRISPR/Cas9-guided integration of the *PgDS*, *PgPPDS*, and *PgCPR* genes, sgRNAs targeting downstream of *TEF2*, upstream of *GLK1*, and upstream of *RPS17B* were designed using an online tool (https://www.atum.bio/eCommerce/cas9/input, Accessed 2 July 2018). All integration sites in this study were previously chosen in our laboratory. *TEF2*- and *GLK1*-targeting sgRNAs were cloned into the pRS42K vector, and the *RPS17B*-targeting sgRNA was cloned into the pRS42H vector. The N-terminally truncated *HMG-CoA* reductase 1 (*tHMGR1* gene), with the BamHI and XhoI restriction sites at its 5′- and 3′- ends, respectively, was amplified from yeast genomic DNA and cloned into the pRS426GPD vector. The *EGR1m* gene, in which four lysine residues of the yeast *ERG1* gene had been replaced with arginine residues (K278R, K284R, K311R, and K360R) in a previous study (Foresti et al., 2013), with the BamHI and XhoI restriction sites at its 5′- and 3′- ends, respectively, was synthesized with codon optimization (BIONEER), and was cloned into the pRS424TEF1 vector. For CRISPRi-mediated *ERG7* suppression, *ERG7* promoter-targeting sgRNAs were designed using an online tool (https://lp2.github.io/yeast-crispri/, Accessed 2 July 2018) (Appendix A). To construct the dCas9/sgRNA co-expression plasmids (pTDH3-dCas9-sgRNA1–5), *SNR52*pro-sgRNA(1–5)-*SUP4t*er cassettes, each with the EcoRI and SpeI restriction sites at their 5′- and 3′- ends, respectively, were synthesized (BIONEER) and cloned into pTDH3-dCas9-Mxi1. The plasmids and primers used are listed in Appendix A, respectively.

### 4.3. Engineering S. cerevisiae for PPD Production

*Saccharomyces cerevisiae* strain CEN.PK2-1D was transformed using the Alkali-Cation Yeast Transformation Kit (#112200200, MP Biomedicals, Solon, OH, USA), according to manufacturer’s protocol. The PPD-A1 strain was constructed by integrating the *PgDS*, *PgPPDS*, and *PgCPR* genes downstream of *TEF2*, upstream of *GLK1*, and upstream of *RPS17B*, respectively, in the CEN.PK2-1D strain.

The *GPD*pro-*PgDS*-*CYC1*ter cassette, with the homologous recombination region of the partial *TEF2* site, was amplified from the *PgDS*-carrying pRS426GPD vector, using primer set TEF2-Integ-F/TEF2-Ineg-R. This integration cassette, with plasmids Cas9-NAT and pRS42K-sgRNA (*TEF2*), was co-transformed into strain CEN.PK2-1D, followed by selection on a YPD/ClonNAT/G418 plate. Strains were verified by diagnostic PCR, and colonies containing the desired plasmids and integration cassette were cultivated at 30 °C in YPD/ClonNAT medium for 24 h to remove pRS42K-sgRNA (*TEF2*), producing the CEN.PK2-1D-PgDS-Cas9-NAT strain.

The *GPD*pro-*PgPPDS*-*CYC1*ter cassette, with the homologous recombination region of the partial *GLK1* site, was amplified from the *PgPPDS*-carrying pRS426GPD vector using the primer set GLK1-Integ-F/GLK1-Integ-R. This integration cassette, with the plasmid and pRS42H-sgRNA (*GLK1*), was co-transformed into strain CEN.PK2-1D-Cas9-NAT-PgDS, followed by selection on a YPD/ClonNAT/Hygromycin B plate. Strains were verified with diagnostic PCR, and colonies containing the desired plasmids and integration cassette were cultivated at 30 °C in YPD/ClonNAT medium for 24 h to remove pRS42H-sgRNA (*GLK1*), producing the CEN.PK2-1D-Cas9-NAT-*PgDS*-*PgPPDS* strain.

The *PGK1*pro-*PgCPR*-*CYC1*ter cassette, with the homologous recombination region of the partial *RPS17B* site, was amplified from the *PgCPR*-carrying pRS426PGK1 vector, using the primer set RPS17B-Integ-F/RPS17B-Integ-R. This integration cassette, with plasmid pRS42K-sgRNA (*RPS17B*), was co-transformed into strain CEN.PK2-1D-Cas9-NAT-*PgDS*-*PgPPDS*, followed by selection on a YPD/ClonNAT/G418 plate. Strains were verified by diagnostic PCR, and colonies containing the desired plasmids and integration cassette were cultivated at 30 °C in YPD medium for 24 h to remove Cas9-NAT and pRS42K-sgRNA (*RPS17B*), producing the PPD-A1 strain.

Strain PPD-A2 was constructed by integrating the *tHMG1* gene downstream of the *TCB2* site of PPD-A1. The [*GPD*pro-*tHMG1*-CYC1ter]-*URA3* cassette, with the homologous recombination region of the partial *TCB2* site, was amplified from the *tHMG1*-carrying pRS426GPD vector, using the primer set TCB2-Integ-F/TCB2-Integ-R. This integration cassette was transformed into strain PPD-A1, followed by selection on an SD/-URA plate. Strains were verified by diagnostic PCR, and colonies containing the integration cassette were cultivated at 30 °C in SD/-URA medium, producing the PPD-A2 strain.

Strain PPD-A3 was constructed by integrating the *ERG1m* gene into the *trp1-289* site of PPD-A2. The [*TEF1*pro-*ERG1m*-*CYC1*ter]-*TRP1* cassette, with the homologous recombination region of the partial *TRP1* gene, was amplified from the *ERG1m*-carrying pRS424TEF1 vector, using the primer set TRP1-Integ-F/TRP1-Integ-R. This integration cassette was transformed into strain PPD-A2, followed by selection on an SD/-URA/-TRP plate. Strains were verified by diagnostic PCR, and colonies containing integration cassette were cultivated at 30 °C in SD/-URA/-TRP medium, producing the PPD-A3 strain.

Strains PPD-A3-sgRNA1–5 were constructed by integrating the dCas9 and *ERG7* promoter-targeting sgRNA genes upstream of the *ISR1* site of PPD-A3. The dCas9-[*SNR52*pro-sgRNA-*SUP4*ter]-*LEU2* cassettes, with the homologous recombination region of the partial *ISR1* site, were amplified from p*TDH3*-dCas9-Mxi1-sgRNA1–5 using the primer set ISR1-Integ-F/ISR1-Integ-R. Each integration cassette was transformed into PPD-A3, followed by selection on an SD/-URA/-TRP/-LEU plate. Strains were verified by diagnostic PCR, and colonies containing the integration cassette were cultivated at 30 °C in SD/-URA/-TRP/-LEU medium, producing strains PPD-A3-sgRNA1–5.

### 4.4. Yeast Cultivation and Batch-Fed Fermentation

For shake-flask fermentation, YPD medium was used to cultivate the yeast strains. First, 1 mL of stock cells in 25% glycerol was inoculated into a 250 mL baffled flask (TriForest, Irvine, CA, USA) containing 29 mL of YPD medium and cultivated at 26 °C, with shaking at 220 rpm, to an optical density at 600 nm (OD600) of approximately 6.0 which was measured by a GENESYS 20 visible spectrophotometer (Thermo Scientific, Waltham, MA, USA). Then, 1 mL of seed culture was inoculated into a 250 mL baffled flask containing 29 mL of YPD medium and cultivated at 26 °C, with shaking at 220 rpm, for 48 h. Flask-fermentation results are presented as means with standard deviation based on biological triplicates.

Strains PPD-A3 and PPD-A3-sgRNA4 were used for the production of PPD via batch-fed fermentation in a stirred glass tank 5 L bioreactor (CNS, Daejeon, Korea), with an initial working volume of 2 L YPD medium. Seed culture was prepared in two steps. First, 1 mL of stock cells in 25% glycerol was inoculated into a 250 mL baffled flask containing 29 mL of YPD medium, followed by cultivation at 26 °C and shaking at 220 rpm, to an OD600 of approximately 6.0. Then, 12 mL of the first seed culture was inoculated into two 1 L baffled flasks (Duran) containing 138 mL of YPD medium, and cultivated at 26 °C with shaking at 220 rpm, to an OD600 of approximately 7.0. Then, 300 mL of the second seed culture was inoculated into the 5 L bioreactor containing 1.7 L of YPD medium. Fermentation was carried out at 26 °C with shaking at 300 rpm and air flow at 4 L/min. pH was controlled at 5.5 by automatic addition of 15% ammonium hydroxide (*v*/*v*). To minimize foaming in the bioreactor, 10% Antifoam 204 (*v*/*v*) was used. After the initial glucose was completely consumed, a solution containing 500 g/L glucose, 18.7 g/L KH_2_PO_4_, 6.5 g/L K_2_SO_4_, 0.53 g/L Na_2_SO_4_, 9.75 g/L MgSO_4_·7H_2_O, 10 g/L histidine, 10 g/L leucine, 10 mL/L of trace metal solution, and 12 mL/L of vitamin solution [23], was added to the bioreactor, and the dissolved oxygen level was increased to above 50%. The glucose concentration in the bioreactor was maintained below 10 g/L.

### 4.5. Metabolite Extraction and HPLC Analysis

Yeast cells at a density equivalent to an OD600 of 40 were collected from the flask-fermentation and batch-fed fermentation processes into 2 mL Safe-Lock Eppendorf tubes, and centrifuged at 13,000× *g* for 5 min. The supernatant was discarded, and the collected cells were resuspended in 1 mL of a methanol–acetone mixture (1:1 *v*/*v*). The cells were then lysed using an MM400 homogenizer (Retsch, Germany), according to the manufacturer’s protocol. Samples were then centrifuged at 13,000× *g* for 10 min. The supernatant (30 μL) was then injected into an Agilent 1260 Infinity II HPLC system (Agilent, Santa Clara, CA, USA), with UV detection at 203 nm. Chromatographic separation was conducted using a Prodigy 5 μm ODS-2 LC column (4.6 mm × 150 mm, Phenomenex, Torrance, CA, USA). The mobile phase consisted of water (A) and acetonitrile (B), using a gradient program of 32–65% B at 0–8 min, 65–90% B at 8–12 min, 90% B at 12–20 min, 90–100% B at 20–30 min, 100% B at 30–65 min, 100–32% B at 65–66 min, and 32% B at 66–70 min. The solvent flow rate was 1.0 mL/min and the column temperature was set to 30 °C. Analytical grade squalene, 2,3-oxidosqualene, lanosterol, and ergosterol were purchased from Sigma-Aldrich (Burlington, MA, USA). Dammarenediol-II and PPD were purchased from ChemFaces (Wuhan, China).

### 4.6. RNA Extraction and qRT-PCR

Yeast cells at a density equivalent to an OD600 of 20 were collected from the flask-fermentation process into a 2 mL Safe-Lock Eppendorf tube and centrifuged at 13,000× *g* for 5 min. Total RNA was extracted using the RNeasy Mini Kit (Qiagen, Germany), and cDNA was obtained using M-MLV Reverse Transcriptase (Thermo Scientific, Waltham, MA, USA), both according to the manufacturers’ protocol. The relative mRNA level of each gene was determined using real-time PCR (Bio-Rad, Hercules, CA, USA), using specific primer sets (Appendix A), with ACTIN1 as the reference gene.

## Figures and Tables

**Figure 1 ijms-22-11836-f001:**
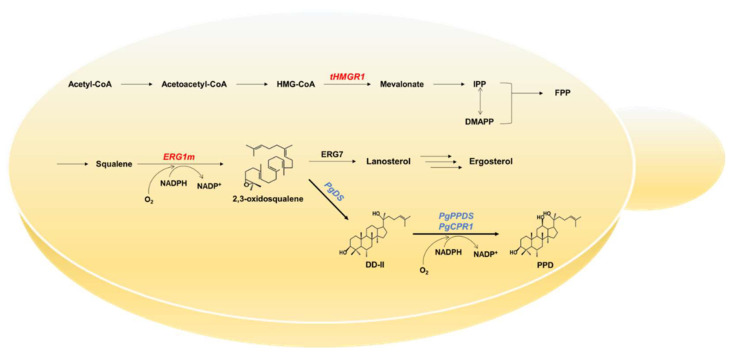
A biosynthetic pathway for protopanaxadiol (PPD) production in the metabolically engineered *Saccharomyces cerevisiae*. Blue: heterologous genes from *Panax ginseng*; red: engineered genes from *S. cerevisiae*. HMG-CoA, 3-hydroxyl-3-methylglutaryl coenzyme A; IPP, isopentenyl pyrophosphate; DMAPP, dimethylallyl pyrophosphate; FPP, farnesyl diphosphate; DD-II, dammarenediol-II; PgDS, *P. ginseng* dammarenediol-II synthase; PgPPDS, *P. ginseng* protopanaxadiol synthase; and PgCPR, *P. ginseng* cytochrome P450 reductase.

**Figure 2 ijms-22-11836-f002:**
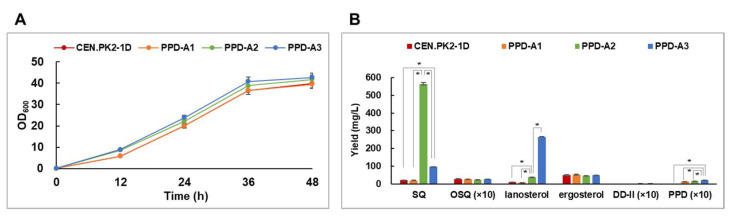
Enhanced production of lanosterol and protopanaxadiol (PPD) in metabolically engineered *Saccharomyces cerevisiae* strains. (**A**) Cell growth curves of engineered strains during flask culture in YPD medium. (**B**) Production of squalene (SQ), 2,3-oxidosqualene (OSQ), lanosterol, ergosterol, dammarenediol-II (DD-II), and PPD in engineered strains at 48 h. Data are presented as means with standard deviation of biological triplicates. Statistical analysis was performed using Student’s *t*-test (*, *p* < 0.05).

**Figure 3 ijms-22-11836-f003:**
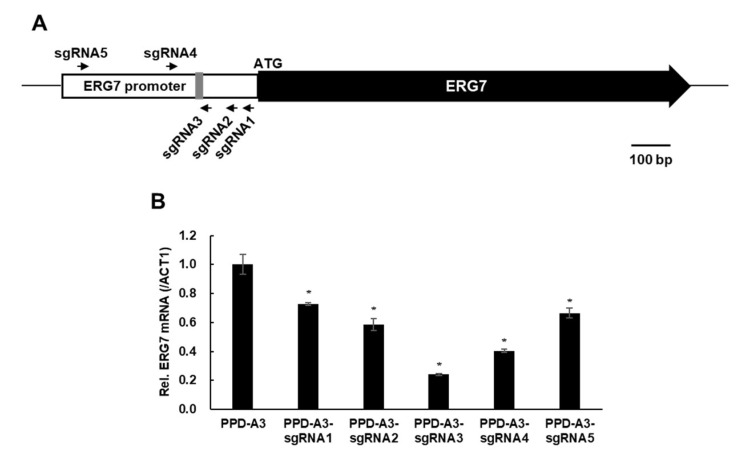
CRISPRi-mediated *ERG7* suppression in the protopanaxadiol (PPD)-A3 strain. (**A**) Schematic of *ERG7* promoter-targeting sgRNAs. The arrowhead of each sgRNA indicates the sgRNA direction. The gray vertical bar in the *ERG7* promoter indicates the TATA-box. (**B**) Relative *ERG7* mRNA levels in engineered strains cultivated for 48 h in YPD medium. Data are presented as means with standard deviation of biological triplicates. Statistical analysis was performed using Student’s *t*-test (*, *p* < 0.05).

**Figure 4 ijms-22-11836-f004:**
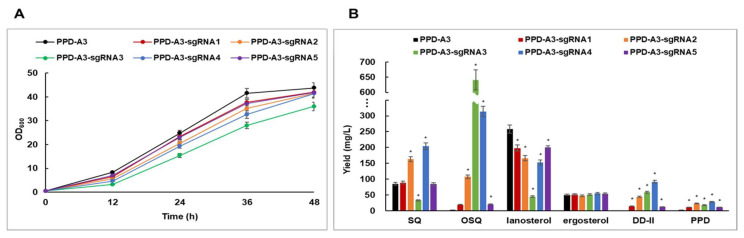
Enhanced production of protopanaxadiol (PPD) by dCas9-sgRNA cassettes in metabolically engineered *Saccharomyces cerevisiae* strains. (**A**) Cell growth curves of engineered strains during flask culture in YPD medium. (**B**) Production of squalene (SQ), 2,3-oxidosqualene (OSQ), lanosterol, ergosterol, dammarenediol-II (DD-II), and PPD in engineered strains at 48 h. Data are presented as means with standard deviation of biological triplicates. Statistical analysis was performed using Student’s *t*-test (*, *p* < 0.05).

**Figure 5 ijms-22-11836-f005:**
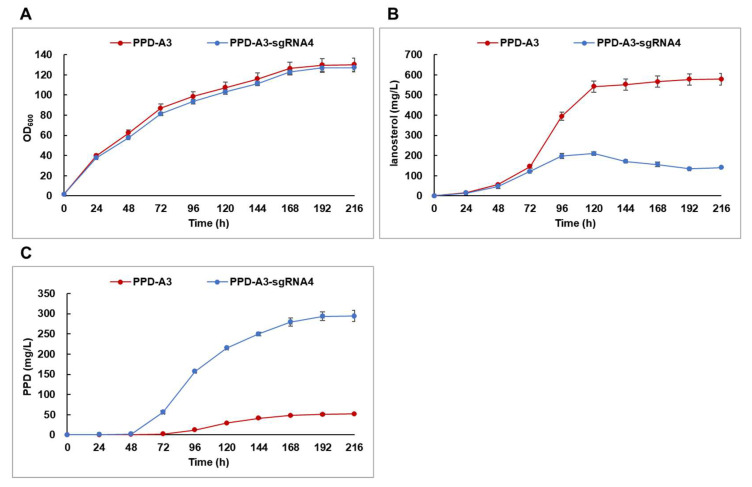
Batch-fed fermentation of protopanaxadiol (PPD)-A3 and PPD-A3-sgRNA4 strains in 5 L fermenter. (**A**) Cell growth curves during batch-fed fermentation. Production of (**B**) lanosterol and (**C**) PPD during batch-fed fermentation.

**Table 1 ijms-22-11836-t001:** Strains used in this study.

	Description	Source
CEN.PK2-1D	*MATα*, *ura3-52*, *trp1-289*, *leu2-3,112*, *his3-Δ1*, *MAL2-8^c^*, *SUC2*	EUROSCARF
PPD-A1	*GPD*pro-*DS*-*CYC1*ter, *GPD*pro-*PgPPDS*-*CYC1*ter, *PGK1*pro-*PgCPR1*-*CYC1*ter cassettes were inserted into CEN.PK2-1D	This study
PPD-A2	*GPD*pro-*tHMGR1*-*CYC1*ter cassette was inserted into PPD-A1	This study
PPD-A3	*TEF1*pro-*ERG1m*-*CYC1*ter cassette was inserted into PPD-A2	This study
PPD-A3-sgRNA1	[*GPD*pro-*dCas9*-*ADH1*ter]-[*SNR52*pro-sgRNA1*_ERG7_*_pro_-*SUP4*ter] cassette was inserted into PPD-A3	This study
PPD-A3-sgRNA2	[*GPD*pro-*dCas9*-*ADH1*ter]-[*SNR52*pro-sgRNA2*_ERG7_*_pro_-*SUP4*ter] cassette was inserted into PPD-A3	This study
PPD-A3-sgRNA3	[*GPD*pro-*dCas9*-*ADH1*ter]-[*SNR52*pro-sgRNA3*_ERG7_*_pro_-*SUP4*ter] cassette was inserted into PPD-A3	This study
PPD-A3-sgRNA4	[*GPD*pro-*dCas9*-*ADH1*ter]-[*SNR52*pro-sgRNA4*_ERG7_*_pro_-*SUP4*ter] cassette was inserted into PPD-A3	This study
PPD-A3-sgRNA5	[*GPD*pro-*dCas9*-*ADH1*ter]-[*SNR52*pro-sgRNA5*_ERG7_*_pro_-*SUP4*ter] cassette was inserted into PPD-A3	This study

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
