# Peer review of "CRISPRi-Guided Metabolic Flux Engineering for Enhanced Protopanaxadiol Production in Saccharomyces cerevisiae"

_ijms, 2021, doi:10.3390/ijms222111836_

Round 1

Reviewer 1 Report

Line 43: correct including as yeast

Figure 2B, Figure 3 and Figure 4: the statistics is missing

Line 301: the composition of YPD and SD is incorrect "YPD medium (10 g/L yeast extract, 20 g/L peptone, and 50 g/L glucose"  this concentration of glucose is not correct, otherwise authors must explain the increase of glucose

Line 305: Escherichia coli cells strain?

Line 301: the growth temperature is needed

Line 184: It would be important to check the viability of the different strains after the ERG7 supression.

Figure 4A: The growth curves are not informative enough, the growth parameters should be calculated in order to compare and perform statistics

Although PPD-A3-sgRNA4 presented an impairment in its growth, authors select the strain because of its metabolite production but they should discuss and analyze the viability of this strain comparing with the parental. 

Reviewer 2 Report

The authors used crispr-based inhibition to suppress the expression of ERG7 and improve the production of their metabolite of interest namely PPD. The manuscript is well-written, and with minor editing could be accepted for publication. Well done the authors.

some issues (all minor)

-ABSTRACT: listing all the potential application for PPD in the first line is quite distracting and would advise the authors to just stick to the "high potency as a pharmaceutical". Rather elaborate in the introduction.

-line 41: the word "very" is a subjective word and should be omitted

-line 43: remove "as"

-line 61, 66, 67, 85, 86 and many other places in the manuscript the gene name is not in italics. Please ensure they are.

-line 67: to repress flux one could also fuse the enzyme to a degradation tag so please cite this paper 10.1016/j.ymben.2016.12.003

-310 the accession numbers of the plasmids are lacking and in general the place names of the manufacturers described in the materials and methods are lacking. For instance "Addgene (USA)" or (MP, USA) line 340 are not sufficient.

-line 316 codon optimization.... for expression in S. cerevisiae (I presume)

-line 395 it's not clear if a spectrophotometer was used and if so which one from Thermo Scientific)

-line 401 what type of 5L bioreactor was used.

-some references are not according to the style of the journal like nr 5, 12, 21, 31 

Round 2

Reviewer 1 Report

Figure 3: the statistics is confused, authors should analyze the differences comparing with the parental strain PPD-A3. The other comparisons are tot needed. 

Figure 4B: Thanks to include the statistics but the test need to be applied in all the  compounds (SQ, OSQ...). It is obvious that this compunds presented significant differences with the control even if they are not important for the authors if the compounds are in the graph they have to be analyzed.

Table S4: the standard deviation (SD) and the statistics is needed

The name of the genes such as ERG7 must be in italic
